# Cold Shock Proteins Mediate Transcription of Ribosomal RNA in *Escherichia coli* Under Cold-Stress Conditions

**DOI:** 10.3390/biom15101387

**Published:** 2025-09-29

**Authors:** Haoxuan Li, Anna Maria Giuliodori, Xu Wang, Shihao Tian, Zitong Su, Claudio O. Gualerzi, Zhe Sun, Mingyue Fei, Dongchang Sun, Hongxia Ma, Chengguang He

**Affiliations:** 1Engineering Research Center, The Chinese Ministry of Education for Bioreactor and Pharmaceutical Development, College of Life Sciences, Jilin Agricultural University, Changchun 130118, China; 13804344532@163.com (H.L.); 15553057121@163.com (X.W.); tianshihao0427@163.com (S.T.); s18043088950@163.com (Z.S.); 005242@jlau.edu.cn (Z.S.); 2School of Biosciences and Veterinary Medicine, University of Camerino, 62032 Camerino, Italy; annamaria.giuliodori@unicam.it (A.M.G.); claudio.gualerzi@unicam.it (C.O.G.); 3College of Biotechnology and Bioengineering, Zhejiang University of Technology, Hangzhou 310014, China; feimingy@163.com (M.F.); sundch@zjut.edu.cn (D.S.)

**Keywords:** cold-shock protein, ribosomal RNA, transcription elongation

## Abstract

*Escherichia coli* displays strong adaptability for growth and reproduction at low temperatures, with ribosome biogenesis being a critical process for its growth in cold environments. The cold-shock proteins (CSPs) encompass a protein family that can assist bacterial growth at low temperatures by acting as molecular chaperones. In this study, we investigated whether CSP CspA, CspE, and CspI affect ribosomal RNA (rRNA) transcription. Deletion of the single genes encoding these proteins had only a very marginal effect on cellular growth at low temperatures, and rRNA synthesis was hardly affected. Double and triple deletion of the genes encoding these proteins resulted in a much stronger phenotype providing evidence that CspA, CspE, and CspI play an essential role in maintaining 16S rRNA synthesis and enabling optimal cellular growth at low temperatures. These findings suggest the existence of efficient backup mechanisms able to compensate for the absence of a single CSP.

## 1. Introduction

Under stress conditions, such as those caused by a sudden temperature decrease, drought, exposure to antibiotics, an osmotic imbalance, and nutrient deficiency, bacteria respond with the expression of proteins belonging to the CSP family, which enable the cells to adapt to the adverse environmental conditions. For instance, lactic acid bacteria exposed to 10 °C for 4 h during mid-exponential phase before being transferred to 30 °C exhibit a 100-fold increase in viability compared to cells grown only at 30 °C.

The large family of CSPs consists of proteins constituted by 69–74 amino acids exhibiting high sequence homology [1,2]. However, not all CSPs are induced by cold stress. Indeed, among the various CSPs, only CspA, CspB, CspE, CspG, and CspI are directly involved in cold stress in *E. coli* [3], although CspA was found to be one of the most abundant proteins present during the early growth of cells not subjected to stress conditions [4,5]. The CspA protein contains RNP1 and RNP2, two highly conserved RNA-binding motifs [6], which represent the main structural elements involved in binding to DNA and RNA [7]. Sequence alignments show that these proteins share high homology with the conserved domain of eukaryotic nucleic acid-binding Y-box transcription factors suggesting that the bacterial CSPs may also act at the transcriptional level [8]. Indeed, it has been shown that protein CspA can act as a transcriptional activator of cold-shock genes in response to cold shock, stimulating the expression of nucleoid protein H-NS [9] and gyrA [10] by binding to the 5′-ATTGG-3′ motif present in the gene promoter regions. However, the CSPs are not always transcriptional activators but can also act as negative regulators, as in the case of CspE, which downregulates cspA transcription [11]. Furthermore, CspA and its homologous proteins CspE and CspC can act as transcription antiterminators, and their overexpression at 37 °C induces the transcription of nusA, infB, rbfA, and pnp, which are located downstream of multiple transcription terminators present in the metY–rpsO operon [12]. It was postulated that CSPs mediate antitermination via RNA-binding, potentially by destabilizing cold-stabilized hairpins or suppressing backtracking [13]. This hypothesis drew support from nascent RNA interaction data [14], though antitermination effects remained concentration-dependent [15]. CSPs have been shown to affect mRNA secondary structures, correcting misfolded mRNAs [16,17], and regulating protein expression levels. In addition to influencing microbial transcription and translation, CSPs also play a regulatory role in the expression of virulence of pathogenic *E. coli* and *Pseudomonas aeruginosa* [18].

The synthesis of ribosomes is one of the most resource-intensive processes in all living cells, and the number and the activity of ribosomes represent a major growth-limiting factor impacting bacterial growth and reproduction. Regardless of the growth rate, ribosomal activity remains constant, with 85% of the total ribosome pool being active [19]. When bacteria face external pressures, the number of ribosomes must be adjusted to achieve a constant level of utilization, thus avoiding a waste of energy and resources.

In this study we have investigated a possible effect of CSPs CspA, CspE, CspI, CspB, and CspG on rRNA transcription under cold-stress conditions. To this end, we carried out gene knockout experiments and *in vitro* transcription using fluorescence molecular beacons and click chemistry. Our results show that three of these proteins, namely CspA, CspE, and CspI, promote rDNA transcription under cold-stress conditions (15 °C) and that the gene-deletion strains had a lower level of 16S rRNA and a significantly reduced growth at low temperatures compared to the wild-type strain.

## 2. Materials and Methods

### 2.1. Strain and Plasmids

These included the *E. coli* Stellar strain (Takara, Dalian, China), BL21 (DE3) pLysS strain (preserved at the Molecular Genetics Laboratory of the University of Camerino, Camerino, Italy), pETM-11 plasmid (HonorGene, Changsha, China), and pKK-3535 plasmid (generously provided by Professor Noller’s lab at the University of California, Thimman Lab). The pVS10 plasmid (kindly provided by the team of Vladimir Svetlov in New York, NY, USA), pETM11-rpoD plasmid, pKD46, pKD4, pCP20, and pColdIV, and the *E. coli* BW25113 strain are stored at the College of Life Sciences at Jilin Agricultural University. The pUC57 vector was used as a backbone to construct the pUC57-CspA plasmid (used in this study), which contains the cspA gene.

### 2.2. Real-Time Quantitative Reverse Transcription PCR for the Detection of csp mRNAs

For the quantification of *csp* mRNAs, wild-type the *Escherichia coli* BW25113 strain was grown at 37 °C and 15 °C in Luria–Bertani broth, and 5 mL samples were collected when the OD_600_ reached 0.5. RNA samples were prepared using RNAsio Plus reagent (TaKaRa, Dalian, China) followed by the reverse transcription of RNA according to the cDNA reverse transcription kit from PrimeScript^™^ RT reagent Kit, (Takara, Dalian, China). Subsequently, a two-step quantitative polymerase chain reaction (qPCR) analysis was conducted utilizing the SYBR^®^ Premix ExTaq kit (TransGen Biotech, Beijing, China).

### 2.3. Construction of Recombinant Plasmid

Genomic DNA was extracted from *Escherichia coli* BW25113 using a *EasyPure*^®^ Genomic DNA Kit (TransGen Biotech, Beijing, China). Primers were designed based on the genes of *E. coli* BW25113 (see Appendix A), with primers containing two 15 bp nucleotide sequences complementary to the vector. The PrimeSTAR Max DNA Polymerase (TaKaRa) was used for PCR amplification of the cspA, cspB, cspE, cspG, and cspI genes. The pETM11 plasmid available in our laboratory was used as a template for amplification, with the primers as listed in Appendix A. Linear vectors were amplified via PCR, and bands of the desired size were purified or excised from the gel. The concentration of the recovered products was measured using a NanoDrop 1000 spectrophotometer (Thermo Scientific, USA). The linearized pETM11 was fused with the amplified genes via In-Fusion cloning technology (Takara). The successful recombinant plasmids were transformed into CaCl_2_
*E. coli* Stellar competent cells according to the transformation protocol. Transformant colonies were selected on LB solid medium containing Kanamycin upon overnight incubation at 37 °C. Single colonies were picked and streaked onto another plate for further incubation at 37 °C for approximately 12 h. Colony PCR verification was performed using T7 primers, following the reaction conditions outlined in the Premix Taq^TM^ (TaKaRa) manual. Correctly verified single colonies were selected for culture and plasmid extraction and sent for sequencing. The sequencing results were compared with the plasmid gene map to confirm that the target genes did not undergo mutations.

### 2.4. Induced Expression and Purification of Five CSPs

The successfully ligated plasmids were transformed into BL21 (DE3) pLysS competent cells. The transformed cells were spread onto LB solid medium (containing Kanamycin and Chloramphenicol) and incubated at 37 °C for 12–16 h. Colonies from the overnight culture plate were subjected to colony PCR verification using the universal T7 primers.

Colonies that were verified as correct were inoculated into Luria–Bertani (broth) and incubated overnight at 37 °C with shaking at 180 rpm. Subsequently, a 20 mL seed culture was inoculated into 2 L of LB liquid medium (containing Kanamycin (50 ng/mL) and Chloramphenicol (50 ng/mL) and grown until the bacterial density reached approximately OD ≈ 0.6–0.7. Induction was performed by adding IPTG to a final concentration of 0.1 mM, and incubation continued for 4 h before harvesting. Bacterial cells were harvested via centrifugation at 8000 rpm for 10 min at 4 °C. The pellet was washed with physiological saline solution and stored at −80 °C. Bacterial cells were sonicated in a balanced buffer solution I (25 mM Tris-HCl pH = 7.9, 200 mM NaCl, 10% glycerol, 0.1 M PMSF, 0.1 M benzamidine, and 6 mM beta-mercaptoethanol). Sonication was carried out using a sonicator at 200 W power for 30 s per cycle, repeated 10 times with gentle mixing and cooling on ice for 2 min after each cycle. The sonicated bacterial suspension was centrifuged at 12,000 rpm for 50 min, and the supernatant was collected, filtered through a 0.45 μm filter membrane, and kept for further use.

The protein purification process involved washing a 5 mL Ni-NTA column with 30 mL of equilibration buffer (Buffer Solution I with 25 mM imidazole) at a flow rate of 1 mL/min. Samples were loaded slowly into the column at a controlled flow rate of 0.5 mL/min. After sample loading, washing with wash buffer (Buffer Solution I with 50 mM imidazole) was carried out to remove non-specific proteins. The target protein was eluted using Elution buffer (Buffer Solution I with 300 mM imidazole) The eluted proteins were collected in separate labeled tubes. Subsequently, the five different CSPs were treated with recombinant TEV protease to remove the His tags from the proteins. At the end of the incubation with TEV, the concentration of NaCl was increased to 300 mM and the reaction mixture containing the cleaved proteins was loaded again onto the Ni-NTA column equilibrated in Buffer Solution I. The Csp proteins with no His-tag were recovered in the flow-through. The concentrated protein Csp was finally dialyzed against Storage Buffer (25 mM Tris-HCl pH = 8.0, 100 mM MgCl_2_, 50% glycerol, 1 mM DTT) and stored at −80 °C in small aliquots.

### 2.5. In Vitro Transcription Assay

The target gene, which includes all nucleotide sequences of rRNA, was amplified using the pKK3535 plasmid [11] as a template. The primers are listed in Appendix A. The amplified fragments were recovered using the EasyPure^®^PCR Purification Kit (TransGen Biotech). In vitro transcription requires a pure DNA template containing a promoter and ribonucleotides. A typical reaction mixture (200 µL) consisted of 1× Transcription Buffer (40 mM Tris pH 7.9, 10 mM MgCl_2_, 150 mM KCl, 0.1% BSA, 1 mM DTT, 0.002% Triton), 2 mM each of ATP, GTP, UTP, and CTP, 0.4 U/µL RNase inhibitor, 1.5 µL/tube of *E. coli* RNA polymerase, purified as described [20], ~480 ng DNA template, and H_2_O. CSPs were added at concentrations of 0.2 mg/mL or 1 mg/mL and the incubation was carried out at either 15 °C or 37 °C for 0, 1, 2, 3, 4, and 5 h. To terminate the in vitro transcription reaction, 20 μL of 3 M sodium acetate, 100% absolute ethanol (3 volumes), and 3 μL of glycogen were added. The reaction mixture was incubated at −20 °C for 1 h, followed by centrifugation at 4 °C and 12,000 rcf for 10 min. The supernatant was discarded, and the RNA pellet was washed with 1 mL of 80% ice-cold ethanol. After centrifugation at 4 °C and 12,000 rcf for 10 min, the supernatant was discarded, and the RNA was allowed to air dry. Each tube was then supplemented with 18 μL of M.B. Buffer (1 M KCl, 10 mM MgCl_2_, 100 μM Tris-HCl) and 2 μL of 1.5μM M.B. (Appendix A). The mixture was briefly vortexed and centrifuged. After denaturation at 95 °C for 2 min, annealing was performed at 45 °C for 10 min. The reaction mixture was then transferred to a black 96-well enzyme-linked immunosorbent assay plate, and the fluorescence intensity was measured using a fluorescence microplate reader (TECAN Infinite, Swiss) with excitation at 485 nm and emission at 520 nm wavelengths.

### 2.6. Construction of Mutants

The inactivation of *csp* chromosomal genes in *E. coli* was achieved using amplicons with homology to the targeted genes and the λ Red recombinase system [21].

The complete genome sequence of *Escherichia coli* BW25113 strain was obtained from NCBI. Using this sequence as a template, primers for designing upstream and downstream homologous arms were designed. Simultaneously, primers for designing resistance cassette fragments with FRT sites (FLP recognition target) at both ends were designed using the pKD4 plasmid as a template. The primer sequences are listed in Appendix A, with 21 bp and 20 bp complementary base sequences to the resistance cassette in the three upstream and downstream homologous arms.

The three sets of upstream and downstream homologous arm fragments were amplified, along with the resistance gene. DNA fragments were purified and recovered using a EasyPure^®^PCR Purification Kit. Subsequently, the three sets of upstream and downstream homologous arm fragments were fused with the Kanamycin resistance gene fragment, through fusion PCR. These three parts were mixed in a 1:1:1 equimolar ratio, and the amplification was performed for 30 cycles at 95 °C for 20 s, 55 °C for 20 s, and 72 °C for 40 s, with a final extension at 72 °C for 10 min.

Then, 400 ng of pKD46 plasmid, which encodes the lambda Red genes, was introduced into electrocompetent *E. coli* BW25113 and transferred to a pre-chilled 0.1 cm electroporation cuvette. Electroporation was performed at 2100 V, 200 Ω, and 25 μF. The transformed cells were then transferred to 1 mL of pre-warmed SOC liquid medium at 30 °C and incubated at 30 °C with shaking at 200 rpm for 1 h. After centrifugation, approximately 100 μL of the liquid was retained, and the cells were resuspended and spread evenly on LB agar medium +Amp (100 ng/mL) for incubation at 30 °C for 18 h. Positive transformants were selected and cultured in LB liquid medium +Amp (100 ng/mL).

Subsequently, 1 μg of the targeting fragment was introduced into 100 μL of competent cells via electroporation using the same parameters as before. The transformed cells were cultured and selected on LB agar medium (100 ng/mL Amp+ 50 ng/mL Kan). Positive transformants were then grown in LB liquid medium (100 ng/mL Amp+ 50 ng/mL Kan).

Finally, the cells were inoculated onto LB agar medium + Kan (50 ng/mL) and incubated at 42 °C for 12 h to eliminate the pKD46 plasmid. The pCP20 plasmid, which carries a temperature-sensitive replication system and encodes the FLP recombinase, was then electroporated into the competent cells and selected on LB agar medium supplemented with Amp (100 ng/mL) at 30 °C for 18 h. Positive transformants were then grown on LB agar medium at 42 °C for 12 h for plasmid curing. Single colonies were streaked on LB agar, LB agar (100 ng/mL Amp), and LB agar (50 ng/mL Kan) plates and incubated at 37 °C to select for strains without the resistance gene.

### 2.7. Observation of Bacterial Growth Morphology and Determination of Growth Curve

The mutant strains and wild-type strains were inoculated into 5 mL of Luria–Bertani (broth) and cultured until reaching an OD_600_ of 0.5. Subsequently, different dilutions were prepared, and 3 μL of each dilution was spotted onto LB agar plates. The plates were then placed in an incubator at 37 °C or 15 °C for cultivation.

For the growth curve, the strains were cultured overnight in Luria–Bertani (broth), and the bacterial suspension was adjusted to an OD_600_ of 1.0 using a spectrophotometer. The bacterial suspension was then centrifuged, and the pellet was resuspended in sterile PBS to adjust the OD_600_ to 0.6. The seed culture was inoculated into fresh 20 mL of Luria–Bertani (broth) at a 1/100 dilution and incubated at either 15 °C or 37 °C. Three independent experiments were settled for each strain. Sampling started at 0 h, with subsequent samples taken every 4 h (15 °C) or 30 min (37 °C). At each time point, 100 μL of the culture was transferred to a 96-well plate and topped up to 200 μL with fresh Luria–Bertani (broth). The OD_600_ value was measured using a spectrophotometer. Fresh Luria–Bertani (broth) was used as a negative control.

In addition, the wt, triple deletion, and triple deletion+pUC57-cspA strains were grown in the growth medium supplemented with the translation inhibitor Hygromycin B (20 ng/μL), to functionally validate that the cold-sensitive phenotype arises from insufficient translation capacity.

Plate spot growth tests were also performed based on serial dilutions of the above-mentioned strains on Hygromycin B (20 ng/μL) LB agar plates after 12 h at 37 °C and 72 h 15 °C.

The curve fitting for the growth curves shown in the figures was performed using the Gompertz growth equation from Prism (v. 8.3), modified in the form Y = YM*(Y0/YM)^(exp(−KX)) + C. The growth curve fitting for the triple mutant at 15 °C was instead obtained by modifying the equation shown to Y = YM(Y0/YM)^(exp(−KX)) + C + YD(Y0/YD)^(exp(−KD*(X − X0))), as in this case, we observed the combination of two growth phases, identifiable with two Gompertz growth curves, the second of which begins to significantly contribute to X > X0. Both the confidence intervals (95% CI) and the comparison between the parameter values obtained from the best fitting were calculated using Prism v 8.3. Parameter values significantly different (*p*-values < 0.05) are indicated in the tables associated with the plots.

### 2.8. Quantification of rRNA by RT-qPCR of Ribosomal rRNAs

Both wild-type and knockout strains were cultivated at 15 °C or 37 °C in Luria–Bertani (broth). Total RNA was extracted from 5 mL samples taken at a 0.3, 0.5, and 0.8 OD600 from each culture. RNA extraction was performed using the RNAsio Plus reagent (TaKaRa), while the reverse transcription step was carried out using the cDNA reverse transcription kit (PrimeScript^™^ RT reagent, TaKaRa). Real-time quantitative fluorescence PCR was conducted to quantify 16S RNA (16S-F: AGAGTTGATCGTCAG; 16S-R: GGTTACCTTGTTACGACTT) and the reference mRNA gapA encoding the GAPDH enzyme (F: GAAATTCTTGGGCGAATACA; R: CTTTCACCTCGGAAAAGACG) using the TransScript^®^ Green One-Step qRT-PCR SuperMix (Transgen Biotech) with the Agilent Mx3000P instrument. The qPCR reaction program consisted of 40 cycles: pre-denaturation at 94 °C for 30 s, denaturation at 94 °C for 5 s, and annealing at 60 °C for 30 s. Each experiment was repeated 3 times. Changes in gene expression levels were measured using the 2^-ΔΔCt^ method.

### 2.9. Click Chemistry for Nascent rRNA Capture

The functional groups of azide and biotin are located at both ends of the ligand structure, with a central arrangement of 6 straight chains and 2 polyethylene glycol linkers. This design facilitates effective binding as a ligand between the new RNA and the magnetic bead, overcoming potential steric hindrance issues posed by RNA molecules during conjugation.

In the experimental setup, 10 mL of fresh Luria–Bertani (broth) was introduced into a 25 mL conical flask, followed by the addition of an overnight culture of E. coli BW25113 at a 1:100 dilution. The culture was then incubated with agitation at 160 rpm at 15 °C. Upon reaching specific optical density values (OD600 = 0.3/0.5/0.8), rifampicin was introduced into the bacterial culture at a concentration of 10 μg/mL, followed by continued agitation.

After a 30 min incubation period with rifampicin [22,23], a 10 mL sample of the culture was centrifuged at 3000 RPM at 4 °C. The supernatant was carefully removed, and the bacterial pellet was washed with 10 mL of sterile normal saline solution. Subsequent centrifugation and removal of the supernatant were performed to ensure proper cleaning of the bacterial cells.

Upon addition of 10 mL of fresh Luria–Bertani (broth), 400 μM 5-EU was included to label nascent RNA, with a control group lacking 5-EU. Subsequently, shaking incubation was initiated for 1 h before sampling. The samples were promptly centrifuged to collect bacterial cells, followed by discarding the supernatant and washing free 5-EU with sterile saline solution. The bacterial pellets were then resuspended in 1 mL of Trizol reagent, quick-frozen in liquid nitrogen, and stored at −80 °C.

Total RNA extraction was performed using the RNAsio Plus reagent (TaKaRa). Upon completion of the procedure, dried RNA pellets were resuspended in 44 μL of RNase-free water. DNase I treatment was applied to remove residual genomic DNA, followed by inactivation at 80 °C for 2 min with the addition of 2.5 μL of 0.5 mM EDTA. RNA purification was carried out using sodium acetate and the ethanol precipitation method. The total RNA was resuspended in 25 μL of RNase-free water.

For the biotinylation reaction, a mixture was sequentially added to the total RNA sample (1–5 μg) to prepare a 60 μL reaction, containing 100 mM Tris·HCl (pH 7.5), 10% acetonitrile, 250 mM l-ascorbic acid sodium salt (prepared freshly each time), 0.5 mM biotin-PEG_2_-C6-azide ligand, 100 mM PMDTA, and 1 mM CuSO_4_. The mixture was thoroughly combined and incubated at 45 °C in the dark with shaking at 750 rpm for 30 min. Subsequently, the RNA was purified using an ethanol precipitation method, followed by resuspension in 50 μL of Buffer A. The yield was assessed using a NanoDrop measurement, and the biotinylated RNA samples could be stored at −80 °C for up to one week.

Biotinylated nascent RNA was subjected to magnetic bead selection using the optimized EasySep™ Biotin Positive Selection Kit II (Stem Cell Technologies, Canada) protocol. Total RNA was denatured at 65 °C for 5 min, followed by rapid cooling on ice. Ribonuclease was eliminated, and 1 μL of RNasin Plus RNase inhibitor was added to each sample. The total RNA samples were transferred to a 96-well plate and, after mixing with Selection Cocktail (final concentration 100 μL/mL), incubated for 15 min. RapidSpheres magnetic beads (final concentration 75 μL/mL) were then added to the mixed samples and incubated for 10 min. The biotinylated RNA could now bind to the RapidSpheres. The RapidSpheres were immobilized and fixed at the bottom of the 96-well plate using a magnetic rack (EasyPlate™ EasySep™).

The beads were washed twice with Buffer B (10 mM Tris HCl, pH 7.4; 1 mM EDTA, pH 8.0; and 200 mM NaCl), followed by one wash with Buffer A (10 mM Tris HCl, pH 7.4; 1 mM EDTA, pH 8.0; and 2 M NaCl) to eliminate unbound RNA. The RapidSpheres containing biotinylated nascent RNA was resuspended in 10 μL of Buffer A and collected into Eppendorf tubes after removal from the magnetic rack. The nascent RNA obtained after completing the RapidSpheres magnetic bead selection process were then supplemented with 18 μL of M.B. Buffer and 2 μL of 1.5 μM M.B. Signal 1. The mixture was briefly vortexed and centrifuged. After denaturation at 95 °C for 2 min, annealing was performed at 45 °C for 10 min. The reaction mixture was then transferred to a black 96-well enzyme-linked immunosorbent assay plate, and the fluorescence intensity was measured using a fluorescence microplate reader (TECAN Infinite) with excitation at 485 nm and emission at 520 nm wavelengths.

### 2.10. Electrophoretic Mobility Shift Assay with CspA Protein and rrnB DNA

Two *rrnB* amplicons of increasing length and containing the P1 and P2 promoter region were generated via PCR using primers pKK3535-F and pKKprm-R (fragment of 170 bp) and primers pKK450-F and pKK450-R (fragment of 500 bp). For primer sequences, refer to Appendix A; 100 ng of each of these fragments, or the pKK3535 full plasmid, was incubated with increasing amounts of CspA (2 μM, 5 μM, 10 μM, 25 μM). At the end of the incubation, glycerol blue was added, and the samples were loaded on a 8% PAGE under non-denaturing conditions.

## 3. Results

### 3.1. CspA, CspE, and CspI Promote rRNA Transcription at 15 °C

Real-time PCR was used to determine the expression of *cspA*, *cspB*, *cspE*, *cspG*, and *cspI* mRNAs in *E. coli*. The results (Appendix A) indicate that *cspA* mRNA is expressed at high levels at both 37 °C and 15 °C, whereas the expression of the other mRNAs was very low at 37 °C but significantly increased at 15 °C, consistent with the findings that these mRNAs are abundantly expressed at low temperatures [24].

Expression vectors based on the pETM11 plasmid encoding CspA, CspB, CspE, CspG, and CspI were constructed. Upon transformation into BL21 (DE3) competent cells, these proteins were expressed and purified (Appendix A), as described in the Materials and Methods, and their effects on rRNA transcription at 15 °C were investigated by performing in vitro transcription experiments. These tests were carried out using, as a template, an amplicon produced from the recombinant plasmid pkk3535, which contains the entire *E. coli* rDNA sequence (Figure 1). In these experiments, we made use of a molecular beacon (MB) system consisting of two 40-base-long ssDNA oligonucleotides (MB1 and MB2) containing carboxyfluorescein (FAM) and an IBkFQ quencher at their 5′ and 3′ ends, respectively.

The fluorescence values obtained with either MB1 or MB2 did not significantly differ when the transcription tests were carried out at 37 °C in the absence or in the presence of different concentrations of the CSPs (Figure 2A). However, at 15 °C, the transcription level was very low in the absence of the CSPs but increased significantly in the presence of 0.2 mg/mL each of CspA, CspE, and CspI, whereas the level of rRNA transcription was not affected by CspB and increased only slightly after 3 h in the presence of CspG (Figure 2B). The same result was also obtained after increasing the CspB and CspG concentrations up to 1 mg/mL. Taken together, these results indicate that CspA, CspE, and CspI cause a significant increase in rRNA transcription at 15 °C unlike CspB and CspG, which had minimal or no impact on transcription.

### 3.2. Effect of cspA, cspE, and cspI Deletions on Bacterial Growth and 16S rRNA Levels Detected at 37 °C and 15 °C

In light of the above results, we decided to investigate the influence of CspA, CspE, and CspI on *E. coli* growth at a low temperature. For this purpose, we constructed ∆*cspA*, ∆*cspE*, and ∆*cspI* single-gene-deletion strains as described in the Materials and Methods.

Compared to the wt cells, these single-gene-deletion strains did not display a significantly altered phenotype when their growth was followed at 37 °C (Appendix A) and 15 °C (Appendix A) but showed a very slight reduction of growth observed with ∆*cspA* and ∆*cspI* at the low temperature. rRNA transcription at 37 °C (Appendix A) or the low temperature (Appendix A) was likewise not substantially affected in these deletion mutants, but for the case of the ∆*cspA* strain, it displayed a decrease in the16S rRNA level during the initial growth phase at 15 °C (Appendix A).

Because these results indicate that the absence of a single CSP is not sufficient to elicit a growth defect, probably because of the presence of back-up mechanisms, we decided to construct double-gene-deletion strains.

### 3.3. Effect of cspA/cspE, cspE/cspI, and cspA/cspI Double Gene Deletions on Cell Growth and 16S rRNA Levels

In light of the above finding that the single knockouts of the genes encoding CspA, CspE, and CspI have almost no effect on 16S rRNA transcription, we constructed strains carrying double gene deletions and we compared their growth rates. The growth curves of the wt strain and double mutants obtained at 37 °C are perfectly overlapping (Figure 3A), whereas those obtained at 15 °C (Figure 3B) show some differences, albeit limited. In fact, at this temperature, the double mutants *cspE/cspI* and *cspA/cspI* grow at a slightly lower rate than the wt strain, as determined by fitting the data with the Gompertz growth equation, with the growth rate values obtained from the fitting significantly different (*p*-values < 0.05) (Figure 3C). The double mutant *cspA/cspI* shows the most pronounced slowdown, with a growth rate reduction of about 35% compared to that of the wt strain. However, after 50 h at 15 °C, all cultures reach a comparable OD_600_, suggesting that the strains bearing the double gene deletions are able to adapt to the new environment similarly to the wt strain.

Next, we measured the levels of 16S rRNA in both the wt and the double mutants grown up to OD_600_ = 0.3, 0.5, and 0.8 at 37 °C or 15 °C via fluorescence quantitative PCR. The results indicate that upon growth at 37 °C up to OD_600_ = 0.3, there is no significant difference between wt and the double deletion mutants (∆*cspA/cspE*, ∆*cspE/cspI* and ∆*cspA/cspI*) (Figure 3D), whereas there are significant differences in cells grown to the same cell density at 15 °C (Figure 3E). Differences were still visible, albeit to a lower extent, in cells that had reached OD_600_ = 0.5 (Figure 3F) but the differences were no longer significant in cultures that had reached OD_600_ = 0.8 (Figure 3G).

These results suggest that an insufficient amount of CSPs in the double deletion strains results in a delayed growth at a low temperature, possibly due to an insufficient level of 16S rDNA transcription.

### 3.4. Growth of a Mutant Strain Bearing a ∆cspA/cspE/cspI Triple Gene Deletion at 15 °C

The phenotypes displayed at a low temperature by the double deletion mutants prompted further investigation of the role played by CspA, CspE, and CspI in allowing cell growth at low temperatures. For this purpose, we constructed a mutant strain bearing deletions of the three genes encoding these proteins (i.e., ∆*cspA/cspE/cspI*). The rescue cspA gene strain(∆*cspA/cspE/cspI* + pUC57-*cspA*) was also constructed. The growth curves of wt, triple deletion, and rescue cspA gene strains were identical at 37 °C. However, the triple mutant attained a substantially lower plateau (OD_600_ = 1.14) compared to wt cells (OD_600_ = 1.5) after 20 h incubation at 15 °C (Figure 4A). This difference was also observed when the experiment was conducted at 25 °C (Figure 4B). In this cold-shock experiment, unlike the others, the strains grew following a peculiar pattern that can be considered as a combination of two successive Gompertz growth curves, the second of which begins to contribute significantly to X > 36 h. In the initial phase, the cells grew at the usual rate observed for growth at 15 °C, while in the second phase (after 36 h), they started to grow faster (both wt and mutant, see table in panel C), although the mutant stopped growing before the wt cells. To verify whether the difference in OD_600_ observed at 15 °C was indeed due to the cold-sensitive phenotype of the triple mutant, serial dilutions (Figure 4C) of the culture types were plated on LB and allowed to grow for 72 h at 15 °C or for 12 h at 37 °C. Also in this case, we observed a lower number of cells for the triple mutant at 15 °C but not at 37 °C, demonstrating that the triple mutant is indeed temperature-sensitive.

The expression of CspA from the pUC57 plasmid in the triple deletion strains restores, although not completely, the wt phenotype at 15 °C (Figure 4A), and this is particularly visible when the strain growth at 15 °C was compared based on serial dilutions (Figure 4C).

The wt, triple deletion, and triple deletion + pUC57-cspA strains were also grown in the medium containing Hygromycin B (20 ng/μL), a translation inhibitor, in order to functionally validate that the cold-sensitive phenotype stems from impaired translation capacity. In the presence of this inhibitor, overall cell growth was reduced in all strains, both in terms of the growth rate and final OD_600_ (Figure 4A). However, the triple mutant displayed a markedly stronger growth defect compared to the wt, such that after 70 h at low temperature, its OD_600_ was only about one-third that of the wt. When Hygromycin was present, the expression of CspA from the pUC57 plasmid provided only limited support for cell growth, most likely because CspA expression itself was impaired by the antibiotic.

The levels of 16S rRNA were measured via fluorescence quantitative PCR in the wt, triple deletion strain, and triple deletion + pUC57-cspA upon reaching OD_600_ = 0.3, 0.5, and 0.8 at either 37 °C or 15 °C. No differences were observed between the cells grown up to OD_600_ = 0.3 at 37 °C° (Figure 5A), whereas significant differences were seen upon growth up to OD_600_ = 0.3 (Figure 5B) and 0.5 (Figure 5C) at 15 °C. However, the differences became minimal when the cultures had reached OD_600_ = 0.8 (Figure 5D). Notably, the expression of CspA in the triple mutant completely restores the 16S rRNA levels observed in the wt strains (Figure 5E,F).

To quantify the amount of nascent RNA present in the cells as a function of time, a novel technique, known as nascent RNA click chemistry, was used. The antibiotic rifampicin was used to block transcription initiation without affecting the completion of nascent RNA chains. After the removal of rifampicin, ethynyl uridine (5-EU) was used to label nascent RNA. Total RNA was extracted from cell cultures that had reached OD_600_ = 0.3, 0.5, and 0.8 at 15 °C. After removing the genomic DNA, the ethynyl-labeled RNA was biotinylated based on click chemistry, and the nascent RNA was isolated using magnetic beads and detected using a fluorescent molecular beacon. The results showed significant differences in the amount of nascent rRNA between wt and the triple-gene-knockout strain at OD_600_ = 0.3 and 0.5, but not at OD_600_ = 0.8 (Figure 6). These findings are consistent with the results of the fluorescence quantitative experiment and further emphasize the crucial role of CSPs in the transcription of 16S rRNA at low temperature. In addition, when cspA was expressed in the triple mutant, the level of the nascent 16S RNA reached that of the wt strain.

### 3.5. EMSA Test of CspA Protein with rrnB rDNA

To determine whether the effect observed, both in vitro and in vivo, on 16S rRNA synthesis could result from a direct interaction between a CSP protein and the promoters driving ribosomal operon transcription, we incubated increasing amounts of CspA with DNA fragments of different lengths containing the rrnB P1 and P2 promoters. Two of these DNA fragments were generated by amplifying the promoter region together with downstream sequences of different sizes, yielding products of 170 and 500 bp, respectively. In addition, the entire plasmid pKK-3535, which carries the full *rrnB* operon, was also incubated with CspA. Following incubation, the mixtures were analyzed via 8% native PAGE to assess the presence of band shifts. As shown in Figure 7, under the conditions tested, no binding of CspA to these DNA molecules was detected.

## 4. Discussion

Ribosome biogenesis accounts for a significant fraction of the cell‘s total energy budget and represents a critical process for bacterial growth. When mesophilic bacteria like *E. coli* experience an abrupt drop in temperature, ribosome synthesis becomes even more critical for cell survival. As a matter of fact, it is well known from very early studies that the cold-sensitive phenotype displayed by a large number of conditional mutants is caused by defects in ribosome synthesis, assembly, and maturation [25,26].

During the cold adaptation phase, which follows a cold stress in bacteria, the expression of a group of cold-shock genes is transiently activated and a select number of proteins is synthesized [26,27]. Thus, it is not surprising that some of the proteins induced by cold stress are involved in ribosome assembly and maturation, such as ribosomal GTPases LepA [28,29], initiation factor IF2 [30,31], and RbfA [32], in addition to proteins that allow initiation of mRNA translation at low temperatures, such as initiation factors IF3 [33] and IF1 [34], or translation elongation, such as RNase R and CspA, which modulate the mRNA structure during cold acclimation [35,36].

Last, but not least, cold shock induces the expression of members of the “cold-shock proteins” family, a group of small, highly conserved DNA/RNA-binding proteins. In addition to the aforementioned CspA, CspB, CspE, CspG, and CspI are also among the gene products for which cellular levels increase more extensively after the cold stress [2,37,38]. In reality, the term “CSPs” defines a family of proteins not necessarily cold-induced. Indeed, CspD is expressed during the stationary phase and CspA, despite having been defined as “the major CSP” [39], is present at extremely high levels during the initial phases of growth at 37 °C under non-stress conditions [4,40].

A relevant question that remains open concerns the role played by the bona fide cold-induced cold-shock proteins during cold acclimation. It has been shown that proteins such as CspA may facilitate mRNA translation at low temperature [41], or, as in the case of CspE and CspC, may act as transcriptional anti-terminators by virtue of their RNA chaperone properties [13]. Furthermore, these proteins may affect transcription directly either as activators of cold-shock genes, as in the case of CspA, which stimulates the expression of nucleoid protein H-NS [6] and GyrA [10], or as negative regulators, as in the case of CspE, which downregulates *cspA* transcription [12].

The synthesis and maturation of rRNA do not stop during the cold acclimation in *E. coli*, but continue from all seven *rrn* operons, albeit at very reduced pace and with a greater contribution of the P2 compared to the P1 promoter [42].

In light of these multiple functions of the cold-shock proteins, in this study, we investigated a possible role played by these proteins in rRNA transcription during cold acclimation. Preliminary tests indicated that the presence of CspA, CspB, CspE, CspG, and CspI does not affect rRNA transcription at 37 °C, whereas CspA, CspE, and CspI, but not CspB and CspG, stimulate transcription at 15 °C. For this reason, we focused on the effects of these three proteins and constructed mutant strains of *E. coli* in which the genes encoding them were individually deleted. Growth at 37 °C proved to be unaffected in these strains, and only a very minor growth rate slowdown was observed for the mutants at a low temperature. The 16S rRNA level was found to be slightly reduced only in the ∆*cspA* mutant and only at a cell density corresponding to OD_600_ = 0.3, whereas no effect was observed at higher cell density (i.e., OD_600_ = 0.5). These results indicate that the deletion of single *csp* genes has only a marginal effect on cellular growth at low temperatures and only at the onset of growth and that rRNA synthesis is hardly affected implying the existence of efficient back-up mechanisms able to compensate for the absence of a single CSP.

Unlike the case of the single deletion strains, the deletion of pairs of *csp* genes reduced the growth rate more effectively, with the mutant bearing the ∆*cspA*/∆*cspI* deletion found to be somewhat more affected than both the ∆*cspA*/∆*cspE* and ∆*cspE*/∆*cspI* mutants. In addition to the growth defect, this double deletion mutant displayed a severe reduction in the rRNA levels at 15 °C, with the effect being more pronounced at a low cell density (OD_600_ = 0.3) compared to a higher density (OD_600_ = 0.5), while no effect was observed at OD_600_ = 0.8. These phenotypes of the double-gene-deletion mutants were seen only at a low temperature, whereas no effect was observed at 37 °C.

The deletion of three genes (*cspA, cspE*, and *cspI*) resulted in a more severe phenotype compared to the deletion of pairs of genes. After 20 h of incubation at 15 °C (but not at 37 °C), the growth was strongly reduced, as was the cellular level of 16S rRNA, when the cell density corresponded to OD_600_ = 0.3 and 0.5, but returned to almost wt levels when the cell density reached OD_600_ = 0.8. Also, the amount of nascent 16S rRNA was found to be strongly reduced at 15 °C. This may indicate that the strain gradually reduced the impact of cold-shock protein gene deletion during continuous growth.

Taken together, the present results demonstrate that the CSPs CspA, CspE, and CspI play an important role in ensuring the synthesis of 16S rRNA and allowing optimal cellular growth at low temperatures. The lack of these three proteins and also the lack of two of them causes a clear defect in cellular growth and rRNA synthesis under conditions of cold adaptation, but the intensity of these defects tends to diminish after long incubation times, indicating that the cells somehow manage to compensate for the lack of these proteins. On the other hand, the absence of just one of the CSPs does not result in a severely altered phenotype, indicating the existence of efficient back-up mechanisms whereby the function of one of these proteins can be compensated by the other two, in light of the large structural resemblance between them.

The present data indicate, beyond any doubt, that the absence of Csp proteins during growth at 15 °C affects both rRNA synthesis and cell growth. However, it is remarkable that these two defects may not be interconnected. In fact, our data indicate that in the triple deletion mutant, when the 16S rRNA level is drastically reduced, the growth rate is normal, whereas a severe reduction in the growth rate occurs when the rRNA level is restored to almost wt values (Figure 4, Figure 5 and Figure 6) implying that the growth defect results from a defect of some cellular activity controlled by the Csp proteins and not from a reduction of rRNA synthesis. The occurrence of this phenomenon may be due to the deficiency of CSPs, which can affect the protein translation and membrane remodeling processes required for growth. Due to the need for bacteria to resist various adverse environmental factors and consume a significant amount of energy, the nutrients in the culture medium may be insufficient to support further growth and reproduction over an extended period. This can be attributed to a protective mechanism of the bacterial strain itself. Notably, CspA expression rescues strain growth ability almost completely and helps to restore the wt levels of 16S in the cells in the initial phase of growth at low temperatures.

Our EMSA experiments with DNA fragments containing the rrnB operon promoters and increasing amounts of CspA did not appear to support a direct interaction between CspA and DNA. However, it is possible that this assay is not suitable for detecting such an interaction or that the interaction may occur only when the promoter is opened by RNAP. Alternatively, CSPs might promote transcription by interacting with the transcript rather than with the template DNA.

In the process of protein expression in bioengineering, certain substances often require cold conditions for processing. Plasmid p-cold with the cspA promoter is specifically designed for such scenarios [43]. However, this approach does not effectively enhance the growth rate of engineered bacteria. Experiments have shown that CSPs may influence the growth rate of these strains. To shorten the production time, can this current situation be improved by further modifying the plasmid? In vitro transcription assays have revealed that not all CSPs act at the transcriptional level of rRNA, providing a foundation for the selection of CSPs in follow-up research.

Upon completion of this work, an article appeared showing that during cold acclimation, CspA reactivates termination through its RNA chaperone function [44]. This effect of CspA on transcription termination, in addition to that of transcription initiation described here, emphasizes the polyhedral function of this protein during cold acclimation.

## 5. Conclusions

The present data indicate, beyond any doubt, that the absence of Csp proteins affects both rRNA synthesis and cell growth. However, it is remarkable that these two defects do not appear to be interconnected. In fact, our data indicate that in the triple deletion mutant, when the 16S rRNA level is drastically reduced, the growth rate is normal, whereas a severe reduction in the growth rate occurs when the rRNA level is restored to almost wt values (Figure 4, Figure 5 and Figure 6) implying that the growth defect results from a defect of some cellular activity controlled by the Csp proteins and not from a reduction of rRNA synthesis.

## Figures and Tables

**Figure 1 biomolecules-15-01387-f001:**
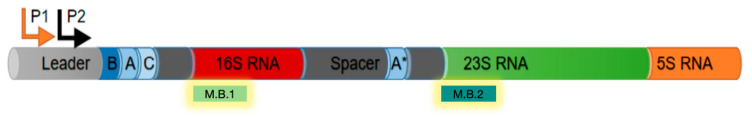
rDNA sequence amplified from the pKK3535 plasmid and sites of 16S and 23S rRNA complementary to molecular beacons (M.B.1 or M.B.2, respectively). The asteric after A is a label to distinguish between the two A region.

**Figure 2 biomolecules-15-01387-f002:**
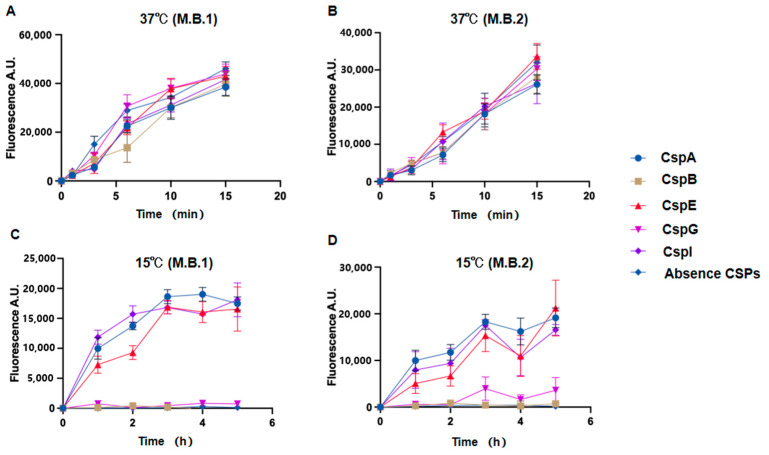
Time courses of rRNA transcription in vitro at 37 °C (**A**,**B**) and 15 °C (**C**,**D**) as detected using M.B.1 (**A**,**C**) or M.B.2 (**B**,**D**) molecular beacons. The reactions were carried out in the presence 0.2 mg/mL of CspA, CspE, CspI, CspB, or CspG or in the absence of CSPs using the indicated molecular beacons. Error bars indicate the standard deviation calculated from triplicate measurements. The background fluorescence was subtracted.

**Figure 3 biomolecules-15-01387-f003:**
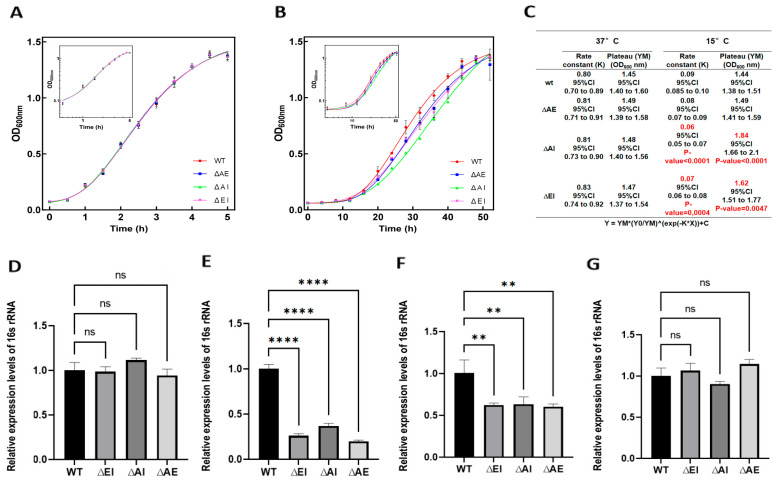
Growth curves and levels of 16S rRNA in double-gene-deletion strains. Growth curves of double csp deletion mutants and wild-type at 37 °C (**A**) and 15 °C (**B**). The table (**C**) shows the main parameters obtained from fitting the experimental points with the Gompertz curve equation as described in the Materials and Methods. Values with a statistically significant difference are indicated in red. The *p*-value was calculated with Prism during the fitting. The insets show the log vs log plots. Error bars indicate the standard deviation calculated from three independent experiments. Quantification of 16S rRNA was carried out via real time PCR on total RNA extracted from wt and from the indicated double-gene-deletion mutants grown at 37 °C (**D**) or 15 °C (**E**–**G**) up to OD_600_ = 0.3 (**D**,**E**) OD_600_ = 0.5 (**F**), and OD_600_ = 0.8 (**G**). The relative expression refers to the ratio between the amount of 16S rRNA in each strain and that of the wild-type strain, measured using the 2^-ΔΔCt^ method. Error bars indicate the standard deviation calculated from triplicate measurements. Data were analyzed using the Anova test (****: *p* < 0.0001; **: *p* < 0.01; ns: *p* > 0.05).

**Figure 4 biomolecules-15-01387-f004:**
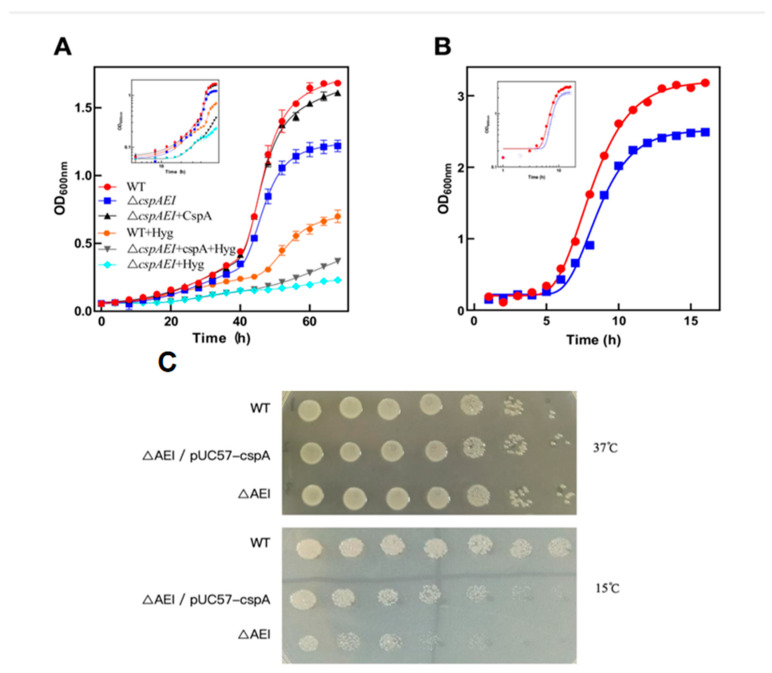
Growth curves and levels of 16S rRNA in single-gene-deletion strains. Growth curves of single *csp* deletion mutants and wild-type at 15 °C (**A**) and 25 °C (**B**). The fitting of the experimental points was carried out using the Gompertz curve equation as described in the Materials and Methods. (**C**) Serial dilutions of the two strains on Hygromycin B (20 ng/μL) LB agar plates after 12 h at 37 °C (upper panel) and 72 h 15 °C (lower panel).

**Figure 5 biomolecules-15-01387-f005:**
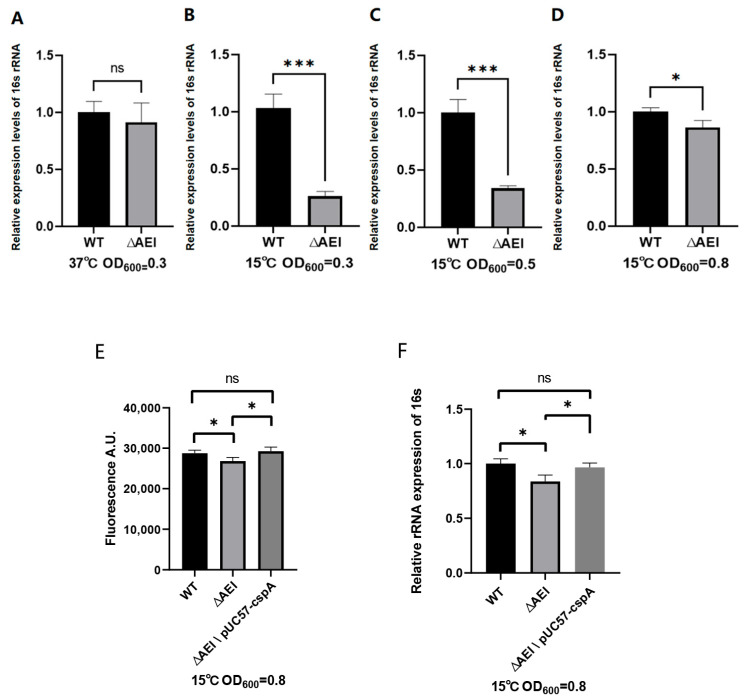
Comparison of the levels of 16S rRNA in wt and triple-gene-deletion cells. The 16S rRNA was quantified via real time PCR based on total RNA extracted when cells had reached OD_600_ = 0.3 at 37 °C (**A**) or 15 °C (**B**). Panels (**C**,**D**) show the 16S rRNA levels detected in cells grown at 15 °C up to OD_600_ = 0.5 and 0.8, respectively. Panels (**E**,**F**) show the 16S rRNA levels detected in cells grown at 15 °C to OD_600_ = 0.3 and 0.8, respectively, when cspA was expressed in the triple mutant. The relative expression refers to the ratio between the amount of 16S rRNA in each strain and that of the wild-type strain, measured using the 2^-ΔΔCt^ method. Error bars indicate the standard deviation calculated from triplicate measurements. Data were analyzed using the Student’s *t*-test (***: *p* < 0.001; *: *p* < 0.05; ns: *p* > 0.05).

**Figure 6 biomolecules-15-01387-f006:**
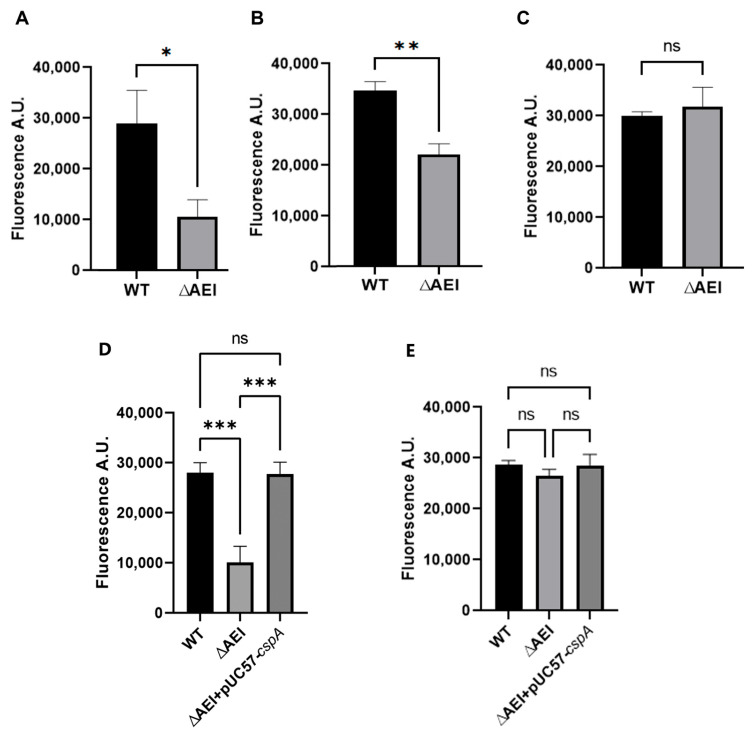
Quantification of nascent RNA via click chemistry. Newly synthesized 16S rRNA was quantified via fluorescence hybridization with MB1 performed on nascent rRNA extracted, as described in the Materials and Methods, from cells grown up to OD_600_ = 0.3 (**A**,**D**) or up to OD_600_ = 0.5 (**B**) and OD_600_ = 0.8 (**C**,**E**) at 15 °C with the indicated strains. Error bars indicate the standard deviation calculated from triplicate measurements. Data were analyzed using the Student’s *t*-test. (***: *p* < 0.001; **: *p* < 0.01; *: *p* < 0.05; ns: *p* > 0.05).

**Figure 7 biomolecules-15-01387-f007:**
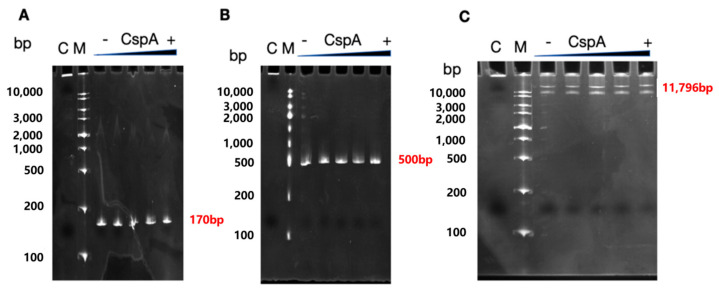
**Electrophoretic mobility shift assay (EMSA) of CspA protein binding to rrnB rDNA fragments.** (**A**) A promoter-proximal fragment of rrnB encompassing the P1 and P2 promoters and the initial 35 bp of the transcribed region, the size is 170 bp (**B**) The 5′-terminal 500 bp fragment of rrnB. (**C**) The full-length of pKK3535 plasmid the size is 11,796 bp. Lane designations: C, positive control (10 μM H-NS protein); M, DNA marker; “-”, no protein control; “+”, increasing concentrations of CspA protein (0 μM, 2 μM, 5 μM, 10 μM, 25 μM). All DNA fragments were amplified via PCR and used in the reaction at the conc. of 100 ng/μL. Refer to Appendix A for primer sequences. (original images can be found in the Appendix A).

## Data Availability

Data is contained within the article.

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
