# Peer review of "Cold Shock Proteins Mediate Transcription of Ribosomal RNA in Escherichia coli Under Cold-Stress Conditions"

_biomolecules, 2025, doi:10.3390/biom15101387_

Round 1
Reviewer 1 Report
Comments and Suggestions for Authors
This manuscript investigates the role of cold shock proteins (CSPs) CspA, CspE, and CspI in ribosomal RNA transcription in E. coli under cold stress conditions. The authors show, through gene knockout experiments and in vitro transcription assays, that these three CSPs promote rDNA transcription at low temperatures, with backup mechanisms compensating for single gene deletions, but significant effects appear in double and triple mutants.
This manuscript presents a systematic investigation of the role of cold shock proteins in rRNA transcription, using a logical progression from single to triple gene knockouts in E. coli. The authors demonstrate functional redundancy among CspA, CspE, and CspI through careful experimental design, showing that single deletions have minimal effects while multiple deletions produce clear phenotypes. The study combines multiple complementary approaches including in vitro transcription assays with molecular beacons, RT-qPCR quantification of rRNA levels, growth curve analyses, and click chemistry for nascent RNA detection. The molecular beacon and click chemistry approaches are particularly well-executed and provide novel technical contributions for studying RNA transcription. The work addresses a genuine knowledge gap regarding CSP function beyond their known roles as RNA chaperones, specifically demonstrating their involvement in rRNA synthesis at low temperatures. The experimental controls are appropriate, statistical analyses are properly conducted using Gompertz curve fitting for growth data, and the authors acknowledge important limitations such as the disconnect between rRNA levels and growth phenotypes in their discussion.
I have following concerns
Major Concerns
- Mechanistic Understanding is Limited
The manuscript demonstrates that CSPs affect rRNA transcription but provides little insight into the molecular mechanism.
Do CSPs directly bind to rRNA promoters or act indirectly?
What is the relationship between transcription enhancement and the known RNA chaperone activity?How do these proteins specifically recognize rRNA genes versus other targets?
Include chromatin immunoprecipitation (ChIP) experiments or electrophoretic mobility shift assays (EMSA) to demonstrate direct DNA binding.
- Disconnect Between rRNA Levels and Growth Phenotype
The authors acknowledge but do not sufficiently explain why growth defects occur when rRNA levels are restored (Figure 4-6). This indicates that CSPs have additional functions beyond rRNA transcription that are essential for cold adaptation. Explore this paradox more thoroughly and consider additional experiments to identify other CSP-dependent processes affecting growth.
- Limited Temperature Range
The study only compares 37°C and 15°C. Testing intermediate temperatures would provide better understanding of the temperature dependence and physiological relevance. Include at least one intermediate temperature (e.g., 25°C) to establish temperature-response relationships.
Minor Concerns
- Writing and Presentation Issues
Several grammatical errors and awkward phrasings throughout (e.g., "allowing optimal cellular growth at low temperatures" - line 22). Some figures are difficult to interpret due to small font sizes
- Technical Issues
Molecular beacon sequences should be validated for specificity. Some experimental details are unclear (e.g., exact rifampicin treatment conditions). Figure legends could be more comprehensive. Supplementary material references are incomplete. Some statistical comparisons seem arbitrary (why not compare all conditions?). The rationale for Gompertz curve fitting could be more clearly explained.
Author Response
Comment 1: [Do CSPs directly bind to rRNA promoters or act indirectly?]
Response 1: Thank you for pointing this out. We agree with this comment. Therefore, CSP protein can act as an anti-terminator to regulate RNA synthesis. CSP protein can bind to newly formed rRNA that has just emerged from RNA polymerase during the synthesis of ribosomal RNA, opening the folded structure of rRNA and unraveling the steric hindrance caused by low temperature. We have found literature to discuss this issue, and we have compiled the discussion into a paragraph and included it in the manuscript.
Comment 2: [What is the relationship between transcription enhancement and the known RNA chaperone activity? How do these proteins specifically recognize rRNA genes versus other targets?
Include chromatin immunoprecipitation (ChIP) experiments or electrophoretic mobility shift assays (EMSA) to demonstrate direct DNA binding.
Disconnect Between rRNA Levels and Growth Phenotype
The authors acknowledge but do not sufficiently explain why growth defects occur when rRNA levels are restored (Figure 4-6). This indicates that CSPs have additional functions beyond rRNA transcription that are essential for cold adaptation. Explore this paradox more thoroughly and consider additional experiments to identify other CSP-dependent processes affecting growth.]
Response 2: Thank you for pointing this out. We agree with this comment. Therefore, we made the EMSA test ( result in Figure S5). We could see from the EMSA test that the CspA protein does not bind to the promoter fragment, nor to the rDNA portion of rrnB, and it does not even bind to the entire promoter combined with the rrnB sequence. Therefore, we conclude that the CspA protein likely binds to certain specially structured rRNA molecules after transcription, rather than interacting with the DNA directly. Previous studies have also indicated that the CspA protein can assist post-transcriptional rRNA and play a role in antitermination.
Therefore, we have included the following content in the manuscript to revisit the phenomenon of RNA transcription rate and bacterial growth asynchrony after cold shock treatment of three knockout strains.
This may indicate that the strain gradually reduced the impact caused by the deletion of the cold shock protein gene during continuous development. However, in Figure 4B, it can also be seen that the gene-deficient strain takes more time to reach od=0.8 compared to the wild-type strain, and ultimately reaches the stable phase, with a much lower density of gene-deficient strains compared to the wild-type strain. The occurrence of this phenomenon may be due to the deficiency of CSPs, which also affects the protein translation and membrane remodeling processes required for self-growth. Due to the need for bacteria to resist various adverse environmental factors and consume a significant amount of energy, the nutrients in the culture medium are insufficient to support further growth and reproduction over a long period of time. This can be attributed to a protective mechanism of the bacterial strain itself.
Comment 3:
Limited Temperature Range
The study only compares 37℃ and 15℃. Testing intermediate temperatures would provide a better understanding of the temperature dependence and physiological relevance. Include at least one intermediate temperature (e.g., 25℃) to establish temperature-response relationships.
Response 2: Thank you for pointing this out. We agree with this comment. Therefore, we have made the growth curve test at 25℃, the result can be found in Figure S3. For the result, we can see: compared with 15℃, both WT and triple deletion strain can reach higher optical density. Also, the triple deletion strain’s growth rate will be lower than WT.
Comment 4:
Minor Concerns
Writing and Presentation Issues
Several grammatical errors and awkward phrasings throughout (e.g., "allowing optimal cellular growth at low temperatures" - line 22). Some figures are difficult to interpret due to small font sizes
Response 4: Thank you to the reviewer for pointing out the errors in the manuscript. We have made the necessary revisions in the manuscript. And adjust the line 22 :
play an essential role in maintaining 16S rRNA synthesis and enabling optimal cellular growth at low temperatures.
Comment 5:
Technical Issues
Molecular beacon sequences should be validated for specificity. Some experimental details are unclear (e.g., exact rifampicin treatment conditions). Figure legends could be more comprehensive. Supplementary material references are incomplete. Some statistical comparisons seem arbitrary (why not compare all conditions?). The rationale for Gompertz curve fitting could be more clearly explained.
Response 5: Thank you for pointing this out. We agree with this comment. Therefore, at the beginning of the experimental design, we conducted dozens of sensitivity and effectiveness tests on 5-EU and the Molecular beacon. All displayed fluorescence values were obtained by removing background fluorescence values from the detected fluorescence values. We have added these experimental data to the supplementary materials. More details could be found in the reference below
- Li, YY. Wang, YT. Wu, ZC. Li, HX. Fei, MY. Sun, DC. Gualerzi, C.. Fabbretti, A. Giuliodori, A.M. Ma, HX. He, CG. et al. Development and Application of Detection Methods for Capture and Transcription Elongation Rate of Bacterial Nascent RNA. *Progress in Biochemistry and Biophysics* **2024**, *51*(9), 2249–2260. https://doi.org/10.16476/j.pibb.2023.047
Rifampicin treatment: Based on the literature review, the transcription speed of E. coli RNA polymerase under various conditions was determined. Through data analysis, we determined that it takes at least 20 minutes to complete the transcription of 6K nt ribosomal RNA under various conditions. Therefore, to ensure any situation, we adopted 30 minutes as the processing time for synchronous transcription initiation of rifampicin.
1. Under standard conditions (37°C, rich medium)
Average elongation rate: 40–80 nt/s
2. Under environmental stress conditions
Low temperature (15–20°C): The rate significantly decreases to 5–20 nt/s.
Nutrient starvation (e.g., carbon limitation): The rate can drop to 10–30 nt/s.
3. Gene-specific variations
Strong promoters (e.g., rrn operon): The rate can reach 80–100 nt/s.
Rifampin reduces the rate to <10 nt/s (Dutta et al., 2011, Nucleic Acids Research).
And we also add the references to the menuscript.
Reference:
22. Bremer H, Dennis PP. (1996)Modulation of Chemical Composition and Other Parameters of the Cell by Growth Rate (ASM Press).
23. Vogel U, et al. (2011) Real-time single-molecule imaging of transcriptional regulation in vivo (Nature Methods, 8:757–760).
24. Proshkin S, et al. (2010) Cooperation between translating ribosomes and RNA polymerase in transcription elongation (Science, 328:504–508).
We have made revisions to several other issues in the manuscript. Thank you!

Reviewer 2 Report
Comments and Suggestions for Authors
Comments to the authors
The manuscript entitled “Cold shock proteins mediate transcription of ribosomal RNA in Escherichia coli under cold stress conditions” presents an original and well-constructed study addressing the role of cold shock proteins CspA, CspE, and CspI in promoting 16S rRNA transcription during cold acclimation. The work combines in vitro transcription assays, gene deletion mutants, RT-qPCR, and an elegant use of click chemistry to label nascent RNA. It offers novel mechanistic insight into how E. coli sustains ribosome biogenesis at low temperatures and adds significant depth to our understanding of cold stress adaptation.
Strengths:
The manuscript addresses a clear and previously unexplored question: whether CSPs directly influence rRNA synthesis under cold stress. The multi-pronged experimental approach, including molecular beacons and nascent RNA labelling, is technically sound and innovative. Results are internally consistent and support the authors' central conclusions.
Suggestions for improvement:
- Complementation experiments
While the gene deletions are convincing, the study would be strengthened by complementing at least one of the deletion strains (e.g., ∆cspA/cspE or the triple mutant) with plasmid-expressed CSP(s). This would confirm that the phenotypes observed—especially reduced rRNA levels and cold sensitivity—are directly due to the loss of these proteins rather than indirect effects or polar mutations.
- Hygromycin sensitivity as a functional readout
While the observed reduction in 16S rRNA levels correlates with slower growth at low temperature, the relationship between ribosome biogenesis and cellular growth is not always linear or immediately apparent. Cells can sometimes partially compensate for impaired ribosome production over time, resulting in eventual catch-up growth, as seen in your mutants.
To functionally validate that the cold-sensitive phenotype arises from insufficient translation capacity, I suggest testing the sensitivity of the mutant strains to a translation elongation inhibitor, such as hygromycin B, particularly at 15 °C. Because hygromycin impairs ribosomal function, strains already compromised in ribosome biogenesis (due to csp deletions) should exhibit enhanced sensitivity compared to wild type. This experiment could amplify phenotypic differences that are otherwise subtle or masked under standard growth conditions. A modest sub-inhibitory dose (e.g., 10–25 µg/mL) could be sufficient to reveal growth differences when cells are stressed both at the transcriptional and translational level.
This would add a useful functional dimension to the molecular and genetic data, further supporting your model that CspA, CspE, and CspI maintain translational capacity during cold acclimation by promoting rRNA synthesis.
- Click chemistry protocol timing and controls
The description of the 5-EU labelling experiment mentions that rifampicin was added before labelling. Since rifampicin inhibits transcription initiation, but not elongation, please clarify the rationale for the timing of 5-EU addition and confirm that labelling reflects newly initiated transcription post-treatment.
For completeness, please mention whether controls such as “no 5-EU” or “no molecular beacon” were performed and yielded negligible signal. Including this information—perhaps in Supplementary Materials—would increase confidence in the specificity of the assay.
Writing and presentation:
The manuscript would benefit from careful language polishing and copyediting. Several sections contain typographical errors (5ºC instead of 5 ºC, or µl and mL), awkward phrasing, and inconsistent terminology (“pa ern” instead of “pattern”, etc.). Improving the flow and clarity would enhance readability and accessibility, especially for an international audience.
Figure improvement:
Please ensure that all growth and expression plots clearly label the identity of the strains (e.g., wild type, ∆cspA, ∆cspA/cspE, etc.) in figure legends and possibly in the graphs themselves. Inset axis do not express to be in logarithmic scale.
Caution on drawing conclusions:
In the Discussion and Conclusion sections, the manuscript makes strong claims that reduced rRNA levels do not cause growth defects, and that Csp proteins must therefore regulate some other cold-related cellular function. While this is an intriguing possibility, the data show only a correlation between recovery of 16S rRNA levels and persistent growth lag in the triple mutant — not a mechanistic uncoupling. Therefore, I recommend tempering this interpretation or explicitly stating it as a hypothesis for future investigation. Without direct measurement of protein synthesis capacity, ribosome abundance, or expression of other cold-response genes, it remains speculative that CSPs control additional unknown pathways affecting growth independently of rRNA synthesis.
Note on limitations:
While this study presents strong correlative evidence that CspA, CspE, and CspI enhance rRNA synthesis during cold stress, it stops short of defining their precise molecular role in the transcription process. The mechanism by which these proteins stimulate rRNA transcription remains unclear: whether they interact directly with RNA polymerase, modify transcription initiation or elongation, or act through changes in rDNA structure or transcriptional antitermination is not addressed. Future studies aimed at uncovering these direct interactions—through footprinting, pull-downs, or promoter binding assays—would be essential to confirm the functional involvement of CSPs in rRNA transcription, beyond correlation.
In addition, while the link between impaired rRNA synthesis and cold-sensitive growth is compelling, functional output such as ribosome abundance or translation efficiency was not directly measured. A brief discussion of these limitations would improve the manuscript’s balance and encourage deeper mechanistic exploration in future work.
Conclusion:
This is a thoughtful and technically robust study that provides novel insights into cold-shock adaptation in E. coli. With modest additional experiments and editorial refinements, the manuscript will make a valuable contribution to the literature on bacterial RNA biology and stress response. I look forward to seeing the revised version.
Author Response
Comments 1: [While the gene deletions are convincing, the study would be strengthened by complementing at least one of the deletion strains (e.g., ∆cspA/cspE or the triple mutant) with plasmid-expressed CSP(s). This would confirm that the phenotypes observed—especially reduced rRNA levels and cold sensitivity—are directly due to the loss of these proteins rather than indirect effects or polar mutations.
Response 1: Thank you for pointing this out. We agree with this comment. Therefore, we have constructed a pUC57-cspA deletion complemented plasmid and transformed it back to the triple deletion strain, then made WT, triple deletion, and complemented strain growth curves together. At last, we make OD=0.8 point rRNA quantity of RT-PCR and molecular beacon analysis. From the result, we can get the complement strain growth ability much higher than triple deletion strain. but couldn’t reach to WT strain. The graph and figures we put in the supplemental data, as Figure S3.]
Comments 2: [Hygromycin sensitivity as a functional readout While the observed reduction in 16S rRNA levels correlates with slower growth at low temperature, the relationship between ribosome biogenesis and cellular growth is not always linear or immediately apparent. Cells can sometimes partially compensate for impaired ribosome production over time, resulting in eventual catch-up growth, as seen in your mutants.]
To functionally validate that the cold-sensitive phenotype arises from insufficient translation capacity, I suggest testing the sensitivity of the mutant strains to a translation elongation inhibitor, such as hygromycin B, particularly at 15 °C. Because hygromycin impairs ribosomal function, strains already compromised in ribosome biogenesis (due to csp deletions) should exhibit enhanced sensitivity compared to wild type. This experiment could amplify phenotypic differences that
are otherwise subtle or masked under standard growth conditions. A modest sub-inhibitory dose (e.g., 10–25 µg/mL) could be sufficient to reveal growth differences when cells are stressed both at the transcriptional and translational level.
This would add a useful functional dimension to the molecular and genetic data, further supporting your model that CspA, CspE, and CspI maintain translational capacity during cold acclimation by promoting rRNA synthesis.]
Response 2: Thank you for pointing this out. We agree with this comment. Therefore, we have used 10 ng/μL Hyg to treat the WT and triple deletion strain and performed the growth curve and growth on plate test. The result could be seen as supplementary Figures S3 and S4. The results show that: low concentration of Hyg could make the growth between WT and triple deletion more significant. Thank you for your suggestion to make the result easier to view!
Comments 3: [Click chemistry protocol timing and controls. The description of the 5-EU labelling experiment mentions that rifampicin was added before labelling. Since rifampicin inhibits transcription initiation, but not elongation, please clarify the rationale for the timing of 5-EU addition and confirm that labelling reflects newly initiated transcription post-treatment.]
Response 3: Thank you for pointing this out. We agree with this comment. Therefore, we have based on the literature review, the transcription speed of E. coli RNA polymerase under various conditions was determined. Through data analysis, we determined that it takes at least 20 minutes to complete the transcription of 6K nt ribosomal RNA under various conditions. Therefore, to ensure any situation, we adopted 30 minutes as the processing time for synchronous transcription initiation of rifampicin.
- Under standard conditions (37°C, rich medium)
Average elongation rate: 40–80 nt/s
- Under environmental stress conditions
Low temperature (15–20°C): The rate significantly decreases to 5–20 nt/s.
Nutrient starvation (e.g., carbon limitation): The rate can drop to 10–30 nt/s.
- Gene-specific variations
Strong promoters (e.g., rrn operon): The rate can reach 80–100 nt/s.
Rifampin reduces the rate to <10 nt/s (Dutta et al., 2011, Nucleic Acids Research).
And we also add the references to the menuscript.
Reference:
- Bremer H, Dennis PP. (1996)Modulation of Chemical Composition and Other Parameters of the Cell by Growth Rate (ASM Press).
- Vogel U, et al. (2011) Real-time single-molecule imaging of transcriptional regulation in vivo (Nature Methods, 8:757–760).
- Proshkin S, et al. (2010) Cooperation between translating ribosomes and RNA polymerase in transcription elongation (Science, 328:504–508).
Comments 4: [For completeness, please mention whether controls such as “no 5-EU” or “no molecular beacon” were performed and yielded negligible signal. Including this information—perhaps in Supplementary Materials—would increase confidence in the specificity of the assay.]
Response 4: Thank you for pointing this out. We agree with this comment. Therefore, we think that at the beginning of the experimental design, we conducted dozens of sensitivity and effectiveness tests on 5-EU and Molecular beacon. All displayed fluorescence values were obtained by removing background fluorescence values from the detected fluorescence values. We have added these experimental data to the supplementary materials. More details could be found in the reference below
- Li, YY. Wang, YT. Wu, ZC. Li, HX. Fei, MY. Sun, DC. Gualerzi, C.. Fabbretti, A. Giuliodori, A.M. Ma, HX. He, et al. Development and Application of Detection Methods for Capture and Transcription Elongation Rate of Bacterial Nascent RNA. *Progress in Biochemistry and Biophysics* **2024**, *51*(9), 2249–2260. https://doi.org/10.16476/j.pibb.2023.047
Writing and presentation:
Comments 5: [The manuscript would benefit from careful language polishing and copyediting. Several sections contain typographical errors (5ºC instead of 5 ºC, or µl and mL), awkward phrasing, and inconsistent terminology (“pa ern” instead of “pattern”, etc.). Improving the flow and clarity would enhance readability and accessibility, especially for an international audience.]
Response 5:Thank you to the reviewer for pointing out the errors in the manuscript. We have made the necessary revisions in the manuscript.
Comments 6: [Please ensure that all growth and expression plots clearly label the identity of the strains (e.g., wild type, ∆cspA, ∆cspA/cspE, etc.) in figure legends and possibly in the graphs themselves. The inset axis does not express in a logarithmic scale.]
Response 6: Thank you to the reviewer for pointing out the errors in the manuscript. We have made the necessary revisions in the manuscript.

Reviewer 3 Report
Comments and Suggestions for Authors
- The authors mentioned that CspA, CspE, and CspI are required for optimal rRNA transcription and growth under cold stress. However, it remains unclear whether the reduction in 16S rRNA is the direct cause of impaired growth or if both are downstream effects of broader stress adaptation failure. Please strengthen the causative link between Csp function and rRNA transcription, possibly through rescue experiments. You can use overexpressing one Csp in double/triple mutants.
- The study shows that CspA, CspE, and CspI promote rRNA transcription but does not explore how this is mechanistically achieved—whether through interaction with RNA polymerase, rDNA promoter elements, or stabilization of transcripts. You can include co-immunoprecipitation or EMSA data, or at least discuss possible models/mechanisms in more depth in the Discussion.
- The triple mutant shows reduced rRNA at OD600 = 0.3 and 0.5 but has a more profound growth defect when rRNA levels appear restored (OD600 = 0.8). This contradicts the hypothesis that impaired rRNA synthesis is the primary cause of growth arrest. You can reframe the conclusions to consider that Csp proteins may affect other aspects of cold adaptation beyond rRNA synthesis, such as translation or membrane remodeling.
- Some figures report p-values, others (especially early PCR data or growth assays) are missing statistical annotations or replicate descriptions. Please ensure all quantitative data (e.g., Fig. 2–6) include error bars, number of replicates, and significance levels where relevant. Also, clarify which tests were used (e.g., ANOVA, t-test).
- The manuscript repeats several phrases (e.g., "cold shock proteins", "16S rRNA synthesis", "growth at 15°C") without adding new information. The Discussion section, in particular, could be condensed. Please streamline the text to improve readability. Avoid reiterating known background multiple times unless it connects to your new findings.
Author Response
Comments 1: [The authors mentioned that CspA, CspE, and CspI are required for optimal rRNA transcription and growth under cold stress. However, it remains unclear whether the reduction in 16S rRNA is the direct cause of impaired growth or if both are downstream effects of broader stress adaptation failure. Please strengthen the causative link between Csp function and rRNA transcription, possibly through rescue experiments. You can use overexpressing one Csp in double/triple mutants.]
Response 1:Thank you for pointing this out. We agree with this comment. Therefore, we have constructed a pUC57-cspA deletion complemented plasmid and transformed it back to the triple deletion strain, then made WT, triple deletion, and complemented strain growth curves together. At last, we make OD=0.8 point rRNA quantity of RT-PCR and molecular beacon analysis. From the result, we can get the complement strain growth ability much higher than triple deletion strain. but couldn’t reach to WT strain. The graph and figures we put in the supplemental data, as Figure S3.]
Comments 2: [The study shows that CspA, CspE, and CspI promote rRNA transcription but does not explore how this is mechanistically achieved—whether through interaction with RNA polymerase, rDNA promoter elements, or stabilization of transcripts. You can include co-immunoprecipitation or EMSA data, or at least discuss possible models/mechanisms in more depth in the Discussion.]
Response 2: Thank you for pointing this out. We agree with this comment. Therefore, CSP protein can act as an anti-terminator to regulate RNA synthesis. CSP protein can bind to newly formed rRNA that has just emerged from RNA polymerase during the synthesis of ribosomal RNA, opening the folded structure of rRNA and unraveling the steric hindrance caused by low temperature. We have found literature to discuss this issue, and we have compiled the discussion into a paragraph and included it in the manuscript.
Comments 3: [The triple mutant shows reduced rRNA at OD600 = 0.3 and 0.5 but has a more profound growth defect when rRNA levels appear restored (OD600 = 0.8). This contradicts the hypothesis that impaired rRNA synthesis is the primary cause of growth arrest. You can reframe the conclusions to consider that Csp proteins may affect other aspects of cold adaptation beyond rRNA synthesis, such as translation or membrane remodeling.]
Response 2: Thank you for pointing this out. We agree with this comment. Therefore, we have included the following content in the manuscript to revisit the phenomenon of RNA transcription rate and bacterial growth asynchrony after cold shock treatment of three knockout strains.
This may indicate that the strain gradually reduced the impact caused by the deletion of the cold shock protein gene during continuous development. However, in Figure 4B, it can also be seen that the gene-deficient strain takes more time to reach od=0.8 compared to the wild-type strain, and ultimately reaches the stable phase, with a much lower density of gene-deficient strains compared to the wild-type strain. The occurrence of this phenomenon may be due to the deficiency of CSPs, which also affects the protein translation and membrane remodeling processes required for self-growth. Due to the need for bacteria to resist various adverse environmental factors and consume a significant amount of energy, the nutrients in the culture medium are insufficient to support further growth and reproduction over a long period of time. This can be attributed to a protective mechanism of the bacterial strain itself.
Comments 4: [Some figures report p-values, others (especially early PCR data or growth assays) are missing statistical annotations or replicate descriptions. Please ensure all quantitative data (e.g., Fig. 2–6) include error bars, number of replicates, and significance levels where relevant. Also, clarify which tests were used (e.g., ANOVA, t-test).]
Response 2: Thank you for pointing this out. We agree with this comment. Therefore, we have made the necessary revisions.
Comments 5: [The manuscript repeats several phrases (e.g., "cold shock proteins", "16S rRNA synthesis", "growth at 15°C") without adding new information. The Discussion section, in particular, could be condensed. Please streamline the text to improve readability. Avoid reiterating known background multiple times unless it connects to your new findings.
Answer 5: Thank you to the reviewer for pointing out the errors in the manuscript. We have made the necessary revisions.

Round 2
Reviewer 1 Report
Comments and Suggestions for Authors
Manuscript Number: biomolecules-3793079 (V2)
I would like to thank the authors for their thoughtful revision and careful attention to my comments. Overall, the manuscript has been significantly improved. The addition of new experiments, expanded discussion, and clearer figure legends strengthen the work and make it more convincing. The inclusion of EMSA and intermediate temperature growth data are especially valuable additions. Good work on addressing the primary concerns.
That said, a few issues remain and should be addressed before final acceptance:
Firstly, although additional details and a cited reference were provided, there is no clear experimental validation of the molecular beacon’s sequence specificity. Including a simple control, such as a mismatch beacon or a non-target transcript, would enhance confidence in the method. Secondly, the overall writing has improved, and many figure legends are clearer; however, several grammatical and typographical errors remain, especially in the supplementary material (for example, the EMSA legend contains multiple errors). A thorough language edit is still necessary. Thirdly, the supplementary section still has placeholder text (“Supplementary Materials” boilerplate). Please complete this section with a proper list of figures, tables, and referenced data to ensure clarity and completeness.
The authors have successfully addressed the main scientific concerns and strengthened the manuscript considerably. Once the above minor but important issues are corrected, I believe the manuscript will be ready for publication.
Author Response
Comments 1: [Firstly, although additional details and a cited reference were provided, there is no clear experimental validation of the molecular beacon’s sequence specificity. Including a simple control, such as a mismatch beacon or a non-target transcript, would enhance confidence in the method. ]
Response 1: [We agree with this comment. In order to clarify the specificity of the molecular beacon sequence, we conducted the following experiments and included the results in a supplementary file.
- Melting curves of Molecular beacon(Figure: Melting curves of Molecular beacon)
- Fluorescence emitted by M.B. after incubation with the indicated concentration of complementary target oligonucleotides (Figure S4)
- Molecular beacon’s specificity and sensitivity test. (Figure S5)]
Due to the figure publised in a Chinese language article, so the (Fingure Melting curves of Molecular beacon) I can only pervide in here.

Figure. Melting curves of Molecular beacon Melting curves of Molecular beacon 1 (light green),2 (orange), 3 (cyano) alone or in the presence of a complementary target oligo. The pink, yellow, and red traces refer to another unrelated molecular beacon. The black trace is a control of a FAM-labeled oligonucleotide without a quencher.
The melting curve recorded in a spectrofluorometric thermal cycler at 488 nm from 15°C to 80°C in steps of 1°C (Fig. S7) demonstrates that the fluorescence emission of Molecular beacon 1 (light green),2 (orange), 3 (cyano) are minimal at temperatures < 50°C, in the absence of an excess of the target oligonucleotide with a perfectly complementary sequence, respectively. However, as temperature increases, also fluorescence increases due to the denaturation of the secondary structure. In the presence of the target complementary oligo, fluorescence of Molecular beacon M.B.1+ target oligo(pink), M.B.2+ target oligo (red) and M.B.3 + target oligo( yellow) increases immediately and reaches its maximum at about 45°C, due to the formation of target-M.B. hybrids which take apart the quencher from the fluorophore. However, with increasing temperatures, fluorescence decreases again because the fraction of hybrids is reduced. Finally, when temperature is > 65°C, the Molecular beacon is denatured even in the presence of the complementary target.]
Comments 2: [Secondly, the overall writing has improved, and many figure legends are clearer; however, several grammatical and typographical errors remain, especially in the supplementary material (for example, the EMSA legend contains multiple errors). A thorough language edit is still necessary. ]
Response 2: [We agree with this comment, and we clearly checked the EMSA legend and changed this figure from supplementary data into the manuscript file. And correct other errors. And also, we put growth curve and plate growth spot figures from supplementary data into the Manuscript file as Figure 4. In addition, we collected the 16S rRNA, which was quantified by real-time PCR data of the new growth curve put in the Manuscript as Figure 5E and F.]
Comments 3: [Thirdly, the supplementary section still has placeholder text (“Supplementary Materials” boilerplate). Please complete this section with a proper list of figures, tables, and referenced data to ensure clarity and completeness.
Response 3: [We agree with this comment, and we reorganized the supplemental data file.]
Thank you very much for your comments. This makes the Manuscript much better!
He Chengguang
25 Sep 2025

Reviewer 3 Report
Comments and Suggestions for Authors
My concern has been addressed.
Author Response
Thanks for your comments. The manuscript has much improved after addressing these comments!